# Spatial-Temporal Evolution and Driving Forces of Drying Trends on the Qinghai-Tibet Plateau Based on Geomorphological Division

**DOI:** 10.3390/ijerph19137909

**Published:** 2022-06-28

**Authors:** Yi Liu, Zhongyun Ni, Yinbing Zhao, Guoli Zhou, Yuhao Luo, Shuai Li, Dong Wang, Shaowen Zhang

**Affiliations:** 1College of Tourism and Urban-Rural Planning, Chengdu University of Technology, Chengdu 610059, China; nica6611@gmail.com (Y.L.); zhaoyinbing06@cdut.cn (Y.Z.); zhouguoli@stu.cdut.edu.cn (G.Z.); 2020050872@stu.cdut.edu.cn (Y.L.); shuailee.cn@gmail.com (S.L.); 2College of Earth Sciences, Chengdu University of Technology, Chengdu 610059, China; wdong6985@163.com (D.W.); 2021020918@stu.cdut.edu.cn (S.Z.); 3School of Geography, Archaeology & Irish Studies, National University of Ireland, Galway (NUIG), H91 CF50 Galway, Ireland; 4Human Geography Research Center of Qinghai Tibet Plateau and Its Eastern Margin, Chengdu 610059, China

**Keywords:** climate change, Temperature-Vegetation Drought Index (TVDI), random forest, geomorphological division, Sanjiangyuan

## Abstract

The Qinghai–Tibet Plateau (QTP) is a sensor of global climate change and regional human activities, and drought monitoring will help to achieve its ecological protection and sustainable development. In order to effectively control the geospatial scale effect, we divided the study area into eight geomorphological sub-regions, and calculated the Temperature-Vegetation Drought Index (TVDI) of each geomorphological sub-region based on MODIS Normalized Difference Vegetation Index (NDVI) and Land Surface Temperature (LST) data, and synthesized the TVDI of the whole region. We employed partial and multiple correlation analyses to identify the relationship between TVDI and temperature and precipitation. The random forest model was further used to study the driving mechanism of TVDI in each geomorphological division. The results of the study were as follows: (1) From 2000 to 2019, the QTP showed a drought trend, with the most significant drought trend in the central region. The spatial pattern of TVDI changes of QTP was consistent with the gradient changes of precipitation and temperature, both showing a gradual trend from southeast to northwest. (2) There was a risk of drought in the four seasons of the QTP, and the seasonal variation of TVDI was significant, which was characterized by being relatively dry in spring and summer and relatively humid in autumn and winter. (3) Drought in the QTP was mainly driven by natural factors, supplemented by human factors. The driving effect of temperature and precipitation factors on TVDI was stable and significant, which mainly determined the spatial distribution and variation of TVDI of the QTP. Geomorphological factors led to regional intensification and local differentiation effects of drought, especially in high mountains, flat slopes, sunny slopes and other places, which had a more significant impact on TVDI. Human activities had local point-like and linear impacts, and grass-land and cultivated land that were closely related to the relatively high impacts on TVDI of human grazing and farming activities. In view of the spatial-temporal patterns of change in TVDI in the study area, it is important to strengthen the monitoring and early warning of changes in natural factors, optimize the spatial distribution of human activities, and scientifically promote ecological protection and restoration.

## 1. Introduction

Drought represents a natural disaster bearing a wide range of impacts that often occur at broad spatial scales and can last for a long time. Not only does drought seriously affect the growth of vegetation, the water cycle, and human life, but also destroys the stability of the ecological environment and restricts the development of the social economy. However, drought is difficult to monitor in time since the occurrence of drought is a slow and dynamic process. This process begins with the reduction of precipitation, followed by the loss of soil moisture and the rise of surface temperature, which eventually leads to vegetation stress and a reduction in productivity [1]. In recent years, many researchers have proposed drought indices or improved existing indexes, such as the Vegetation Condition Index (VCI) [2,3,4], Normalized Difference Water Index (NDWI) [5], Vegetation Health Index (VHI) [6], and Drought Frequency Index (DFI) [7], which have been used to monitor the onset, duration, and intensity of drought and to explore the regional applicability of drought indicators. Among them, the most commonly used drought indices were the Standardized Precipitation Index (SPI) [8] and the Palmer Drought Severity Index (PDSI) [9]. SPI is based on the cumulative probability of precipitation at different time scales and was developed to reflect the impact of changes in water resources on groundwater and agriculture during the study period, thereby revealing the degree of drought in the study area [10], which is often used to capture the drought situation at multiple scales [11]. PDSI is a comprehensive index that uses precipitation and air temperature data to evaluate drought status, and is widely employed to study agricultural drought [12]. However, SPI only uses precipitation data and ignores other drought variables, and thus cannot fully capture the degree of drought in the study area [13]. PDSI has a fixed time scale and cannot be used to evaluate the drought at long time scales [14]. The index may lag behind emerging drought conditions and may not capture drought conditions in mountainous areas with complex climate [15].

Therefore, more comprehensive indicators need to be considered. Using the correlation between the surface temperature and a vegetation index [16], Carlson et al. [17] proposed the Different Water Index (DWI), which uses the Soil–Vegetation–Atmosphere-Transfer (SVAT) model to estimate and describe the trapezoid feature of the characteristic space composed of vegetation cover and surface soil water content. Sandholt et al. [18] proposed a simplified surface drought index, Temperature-Vegetation Drought Index (TVDI), based on temperature, radiation balance, and other factors. TVDI is an index which can monitor the degree of drought in a region by establishing a model based on the relationship between a vegetation index and a surface temperature index. This method only needs to use remote sensing images to monitor the severity of drought, which is beneficial to study the spatial and temporal characteristics of regional drought. Compared with the Standardized Precipitation Index (SPI) and the Standardized Precipitation Evapotranspiration Index (SPEI), TVDI is relatively simpler and has higher accuracy [19]. In addition, a conspicuous feature of TVDI is that it correlates negatively with surface soil moisture [20,21]. Thus, this method can be used to detect changes in soil moisture [22] and drought conditions [23,24]. Chen et al. [25] demonstrated that TVDI can effectively monitor soil water dynamics during the growing season in the Huang-Huai-Hai Plain. Chen et al. [26] combined the TVDI and the VHI method to monitor drought in Central America and assessed the affected agricultural areas.

TVDI is commonly used to study areas across the entire range of surface moisture content, from dry to wet, and from bare soil areas with no vegetation cover to areas completely covered with vegetation [18]. However, affected by factors such as changes in vegetation cover and complex and variable terrain, the edges of the dry and wet fitted by TVDI are unstable, and TVDI cannot clearly express the spatial edge in a theoretical sense [27]. At a large scale, changes in geomorphological types provide precipitation, air temperature, evapotranspiration, and other factors with spatial heterogeneity, and then control the differences in regional vegetation spatial distribution and vegetation cover. Finally, these changes in turn have an impact on the regional climate and the ecological environment. Therefore, the uncertainty of soil moisture estimation by TVDI will increase with soil surface heterogeneity [28], and the TVDI method is only suitable for areas with little topographic variability [29]. The differences in geomorphological divisions not only reflect the changes of regional geological tectonic movement characteristics, but also indicate the differences in ecogeographic systems, such as climate, water resource allocation, biology, and soil properties on a regional scale. Geomorphological division may effectively address heterogeneity-related issues that arise when using the TVDI method in dry and wet edges, and can effectively reduce the error and improve the calculation accuracy.

As an important ecological security barrier in China, the Qinghai–Tibet Plateau (QTP) plays an important role in water conservation, biodiversity protection, soil and water conservation, and is a ‘sensor’ of climate change in Asia and the Northern Hemisphere. The QTP has a very significant warming effect in the background of global warming [30]. Elevated temperatures have increased evapotranspiration and drought probability, which has seriously restricted the development of local agriculture and the animal husbandry economy. Moreover, factors such as vegetation cover, the intensity of human activities, and the distribution of plateau lakes and glaciers have affected drought severity in the QTP. Therefore, methods to effectively monitor the spatial-temporal variability of drought in such a large area as the QTP have become the focus of attention.

However, previous studies have shown significant differences in the spatial and seasonal characteristics of drought in the QTP. For example, Wang et al. [31] pointed out that the central QTP has become wetter using SPI and SPEI to study the drought in China from 1961 to 2012. Bin et al. [32] demonstrated that the southeastern QTP is an area prone to drought, and is a high-risk area by using the SPI method and monthly precipitation data from 616 meteorological stations. Wang et al. [33] believed that drought duration, drought severity, and drought frequency decreased in the central QTP from 1950 to 2006. Kai et al. [34] pointed out that there was a significant wet trend in the eastern QTP from 1961 to 2012 using SPEI, SPI, and the Reconnaissance Drought Index (RDI) method. Feng et al. [35] believed that in the future this region would become wetter in spring, more arid in winter, and more arid in summer by using the SPEI-TH (Standardized Precipitation Evapotranspiration Index-Thornthwaite) method after analyzing the meteorological data of stations in the QTP from 1970 to 2017. Wang et al. [36] believed that the degree of drought in the QTP was relieved by wet conditions in spring, slightly wet conditions in summer and autumn, and more arid conditions in winter from 1994 to 2013. Overall, in terms of spatial characteristics, drought severity in the central and southeastern QTP were found to be changing differently, while the drought status in other regions was not clear. In terms of seasonal characteristics, the drought severity in spring, summer, and autumn differed, and drought occurred in winter for both regions.

In terms of the driving factors of drought in the QTP, previous studies focused on the discussion of factors, such as precipitation, temperature, elevation, slope, and aspect, the distribution of lakes on the plateau, human activities, and other factors. For example, precipitation is often used to discuss changes in the degree of drought in the region due to the significant correlation between precipitation and TVDI [37]. Temperature affects regional heat distribution and is one of the important factors affecting drought changes [38]. As for geomorphological factors, elevation produces vertical differences in regional precipitation and temperature [39], and the slope and aspect produce regional differences in precipitation and air temperature [40]. The QTP has a wide area of lakes, glaciers, and permafrost, and thus the melting of glaciers and the freezing and thawing of permafrost affect the seasonal changes of drought in the region [41]. Human activities are also an important external driving force affecting changes in drought [42]. For example, the expansion of urban construction decreases vegetation cover, while water resource utilization for domestic, industrial and mining uses affect groundwater levels and recharge. Due to the different intensities of human activities, differences in land use will lead to differences in drought severity [43]. However, in an area as large as the QTP, changes in drought must be the result of the comprehensive action of many factors, and such research has received little attention.

Random forest is an algorithm that integrates multiple decision trees through the idea of integrated learning [44]. This algorithm is one of the most commonly used and most powerful supervised learning algorithms. It does not require dimensionality reduction and feature selection. It can better process high-dimensional data and solve regression and classification problems. Moreover, it can be trained quickly and implemented easily [45]. The random forest algorithm can obtain estimates that are easier to explain and understand, such as IncMSE and IncNodePurity. IncMSE, which is the increase in mean squared error, indicates the increase of the error estimated by the random forest model relative to the original error after the variable is randomly selected [46]. IncNodePurity, which is the increase in node purity, indicates the degree of influence for a variable on each decision tree node [47]. Both indices can judge the importance of the influencing factor, and the larger the index value, the more important the variable. Moreover, after controlling the interaction between exploratory variables, the individual marginal effect of each factor can also be visualized [48]. However, at present, few researchers have systematically compared the differences and similarities of the relative importance of drought factors in the QTP using the random forest algorithm.

Therefore, the spatial-temporal differences and driving forces of the arid state in the QTP still warrant attention. This paper attempts to: (1) analyze the feasibility of using the TVDI method to study drought in the QTP, (2) clarify the spatial-temporal variation of drought in the QTP, (3) evaluate the main driving forces affecting the QTP and comprehend the specific range of various factors that affect drought. Based on the previous rules of geomorphological division [49], the workflow of our study was to: (1) divide the QTP into eight geomorphological sub-regions; (2) use the TVDI method to calculate the annual TVDI and seasonal TVDI of eight regions from 2000 to 2019; (3) analyze the spatial-temporal variation characteristics of annual and seasonal drought in the QTP, and summarize the variation of TVDI in the study area; (4) use the partial correlation and the multiple correlation analyses to investigate the relationship between air temperature, precipitation, and TVDI; (5) select eight factors in four categories including meteorology, landform, land use and land cover (LULC), and human activities to construct a random forest model, explore the main factors driving the drying trends in each geomorphological sub-division, and rank the contribution of each driving factor. The results of this paper provide a theoretical basis for the division of drought risk in the QTP, the prevention of drought hazards and the sustainable development of regional economy.

## 2. Study Area

The QTP is the highest plateau in the world and the largest plateau in China. It has an important impact on the formation of geographical elements, such as climate and rivers in China. The latitude and longitude of the QTP ranges from 25°59′56″ N to 39°47′35″ N and 73°30′22″ E to 104°39′50″ E (Figure 1), covering six provinces, including the Tibet Autonomous Region and Qinghai Province, the southern parts of the Xinjiang Autonomous Region, the western parts of Sichuan Province, and small portions of Yunnan Province and Gansu Province.

Referring to the previous results of geomorphological division of the QTP [49], we divided the study area into 8 geomorphic areas (Appendix B, Table A2). The topography of the QTP is complex, with high elevation mountains and many basins, canyons, and lakes. The eight factors are shown in Figure 2. The elevation of the QTP is higher in the west and lower in the east. The alpine areas on its edge bear large undulations and low interior. Affected by its own geographical location, the QTP has formed a unique plateau climate zone. The precipitation in QTP decreases from southeast to northwest (Figure 2b). The temperature has a clear latitudinal zonality, decreasing from south to north (Figure 2c). The land cover types in QTP are dominated by grassland (Figure 2h), which is widely distributed in all parts of the QTP and mainly has low to medium canopy cover. Other land cover types include Gobi, bare land, sandy land, saline–alkali land, and other areas with vegetation cover below 5%, which are distributed in the northern part of the QTP. Forests are mainly distributed in the southeast of the QTP due to the influence of precipitation and temperature. Cultivated, urban, and rural built-up lands are mainly distributed in low-lying, warm valleys with abundant water sources.

## 3. Materials and Methods

### 3.1. Materials

#### 3.1.1. MODIS Data

The relationship between NDVI (Normalized Difference Vegetation Index) and surface temperature can be used to calculate TVDI [50] to study the spatial-temporal drought evolution of the QTP. We used NDVI and surface temperature data from 2000 to 2019 from NASA (https://search.earthdata.nasa.gov/, accessed on 1 March 2020). We selected the MOD13A3 product (MOD/Terra monthly vegetation index data product with spatial resolution of 1 km) for NDVI data, and the MOD11A2 product (MOD/Terra 8-day land surface temperature data product with spatial resolution of 1 km) for surface temperature data. The data projection coordinate system was WGS_1984_UTM_Zone_45N. After batch splicing, band extraction, resampling, and projection, the data were processed using the maximum value synthesis method to obtain annual and quarterly NDVI and daytime surface temperature of the eight geomorphological divisions of the QTP. Using the NDVI and surface temperature data, we calculated TVDI. We used Savizky–Golay filtering to eliminate poor quality observations, and the maximum synthesis method was used to eliminate deviation in the data caused by atmospheric interference and cloud cover.

#### 3.1.2. Soil Moisture Data

There is a negative correlation between soil moisture and TVDI, and the relationship between soil moisture at a depth of 10 cm and TVDI is more significant [25]. Therefore, we used the measured 10 cm soil moisture data to verify whether the TVDI method could capture the drought conditions in the QTP. The soil moisture data were obtained from the National Qinghai–Tibet Plateau scientific data center (http://www.tpdc.ac.cn/zh-hans/, accessed on 6 February 2021). Due to the lack of soil moisture measurements at some sampling points, we selected data collected in summer from 2010 to 2016. After we eliminated the missing sampling point data in this period, we calculated the mean value.

#### 3.1.3. Driving Factors

Drought severity in the region is not only affected by the natural environment, but also human activities. Precipitation and temperature are important factors that affect drought variation, and these two factors determine the spatial distribution of drought grade and drought frequency in the region [51]. Geomorphic factors lead to regional differentiation of drought, among which elevation plays a critical role in the spatial-temporal differentiation of drought spread in plateau mountains [52]. Human activities such as the increase in domestic water supplies and the expansion of urban areas can accelerate the spread of drought [53]. Changes in land cover will affect the supply of atmospheric water, thus affecting the severity of regional drought [54]. Lakes, glaciers and other water areas will also affect the seasonal variation of regional drought [41]. So, we adopted the random forest method and selected six driving factors, including precipitation, temperature, elevation, slope, aspect, the Euclidean distance of water surface land, the Euclidean distance of built-up land, and LULC to explore the main driving factors of drought severity change in various geomorphological divisions of the QTP. The source of data for the driving factors is shown in Table 1, and the coordinate system of the data was WGS_1984_UTM_Zone_45N.

The slope and aspect of the geomorphological factors were extracted from the DEM data, and the Euclidean distance of water surface body was used to represent the degree and scope of influence of the water surface body in the QTP, and the human activity factor was described by the Euclidean distance of built-up land. The land use types were combined according to the attribute table, and finally the data of six main land use types, including forest, grassland, cultivated land, water surface body, built-up land were obtained.

### 3.2. Method

#### 3.2.1. TVDI Calculation

TVDI is an index based on the correlation between a vegetation index and a surface temperature index, and serves to reflect drought severity in a region [18]. After constructing the *NDVI-LST* feature space with the same interval of *NDVI* values combined with the maximum composite method and the minimum composite method, the TVDI value can be calculated using Formula (1).
(1)$$TVDI=\frac{LST-LST_{min}}{LST_{max}-LST_{min}}$$
(2)$$LST_{min}=a_{1}+b_{1}\times NDVI$$
(3)$$LST_{max}=a_{2}+b_{2}\times NDVI$$

In Formula (1), *LST* represents the surface temperature of any pixel in the study area. In Formula (2), $LST_{min}$ is the fitted wet-edge function, representing the minimum surface temperature. In Formula (3), $LST_{max}$ is the fitted dry-edge function, representing the maximum surface temperature [18]. $a_{1}$ and $b_{1}$ are the linear fitting coefficients of the wet-edge function, while $a_{2}$ and $b_{2}$ are the linear fitting coefficients of the dry-edge function. The value of TVDI is between 0 to 1, and the higher the value, the drier the area. In areas with obvious geomorphic differences and large scale areas, there may be errors in using the TVDI method to study drought. Therefore, this paper adopts the method of geomorphological divisions to reduce the uncertainty of the results.

#### 3.2.2. Correlation Analysis

Partial correlation analysis was used to analyze the relationship between TVDI and precipitation and temperature [55]. The partial correlation coefficient between temperature, precipitation, and TVDI was obtained using Formula (4).
(4)$$R_{xy,z}=\frac{Rxy-RxzRyz}{\sqrt{\left(1-R_{xz}{}^{2}\right)}\sqrt{\left(1-R_{yz}{}^{2}\right)}}$$

In Formula (4), $R_{xy,\;z}$ means that the dependent variable z is fixed, and the partial correlation coefficient between the independent variables x and y. Rxy, Rxz, and Ryz are the Pearson correlation coefficients of variables x and y, variables x and z, and variables y and z, respectively. If $R_{xy,z}>0$, it means a positive correlation; if $R_{xy,z}<0$, it means a negative correlation. The larger the coefficient, the higher the correlation. The significance test was performed using a *t*-test.

Similarly, in order to study the relationship between TVDI and precipitation and temperature in the QTP, we applied the multiple correlation analysis method. The calculation formula is shown in Formula (5):(5)$$R_{xy,z}=\sqrt{1-\left(1-R_{xy}^{2}\right)\left(1-R_{xy,z}^{2}\right)}$$

In Formula (5), $R_{xy,z}$ is the multiple correlation coefficient between the dependent variable x and the independent variables y and z. The larger the correlation coefficient, the closer the linear correlation between the elements or variables. The correlation was tested by F-test.

The partial correlation coefficient between TVDI and annual precipitation and average annual temperature from 2000 to 2017 was calculated by Formula (5), then the t-test of the partial correlation coefficient and the F-test of the multiple correlation were performed on the calculation results. The statistics that meet the classification conditions were divided into four categories, as shown in Table 2.

In Table 2, $r_{TVDI\;P,T}$ is the partial correlation coefficient between TVDI and annual precipitation, $r_{TVDI\;T,P}$ is the partial correlation coefficient between TVDI and annual average temperature, $R_{TVDI,\;TP}$ represents the multiple correlation coefficient between TVDI and temperature and precipitation, t and F are the statistical values of the *t*-test and F-test, respectively, and t_0.05_ and F_0.05_ are the 0.05 significance levels of the *t*-test and the F-test, respectively.

#### 3.2.3. Linear Trend Analysis

Linear trend analysis is a method of predicting the change trend of a variable by performing linear regression analysis on a variable that changes over time. The linear trend analysis method can be used to analyze the change of each pixel. We applied this method to analyze the changes of TVDI in the QTP in the recent 20 years. Formula (6) was used to calculate the linear trend rate [57].
(6)$$TVDI\_Slope=\frac{n\times \sum_{i=1}^{n}\left(i\times TVDI_{i}\right)-\sum_{i=1}^{n}i\sum_{i=1}^{n}TVDI_{i}}{n\times \sum_{i=1}^{n}i^{2}-{\left(\sum_{i=1}^{n}i\right)}^{2}}$$

In Formula (6), i is the year of 2001, 2002, 2003…2019, n is the length of the time series; $TVDI_{i}$ is the TVDI value of the year i, TVDI_Slope is the regression value of TVDI pixels. TVDI_Slope > 0 indicates that TVDI in the time series is increasing, while TVDI_Slope < 0 means that the TVDI is decreasing.

To test the consistency of the trend of TVDI with time, we used a *t*-test to judge the level of significance of the change. The *p* value was calculated using the obtained TVDI_Slope of change and the results of the *t*-test, and the two significance judgment criteria were a 95% confidence level and a 99% confidence level. According to Table 3, the trend in TVDI was divided into six levels.

#### 3.2.4. Random Forest Algorithm

We selected eight factors: annual precipitation, annual average temperature, elevation, slope, aspect, Euclidean distance of water surface body, Euclidean distance of built-up land, and LULC. These eight factors were resampled to a spatial resolution of 4 km × 4 km in ArcGIS 10.3 software. The degree of influence of these eight factors on TVDI was analyzed using the Random Forest regression (RF) model, which was implemented by calling the “Random Forest” package in R studio 3.6 software (R Studio, Boston, MA, USA). Each group of samples was divided into two subsets, 70% of the training set and 30% of the test set. On the basis of determining the optimal parameters, the fitting calculation was performed to obtain the importance ranking (IncNodePurity) and local relative importance of the driving factors [48]. The larger the IncNodePurity value, the higher the relative importance of the factors and the deeper the influence on the regional TVDI. The analysis of relative importance of the driving factors can reveal the relationship between a certain range of driving factors and TVDI, where the higher the local relative importance value, the higher the contribution to TVDI.

#### 3.2.5. Verification of TVDI

To verify the accuracy of the TVDI method for monitoring drought, the common method is to perform correlation analysis between measured soil moisture data and TVDI. Due to the lack of soil moisture data in some years and some sampling points, we selected summer soil moisture data from 2010 to 2015 to verify the accuracy of TVDI method and compare the advantages and disadvantages between the two TVDI data with and without geomorphological divisions. In Figure 3 and Figure 4, the *y*-axis represents the measured soil moisture content at 10 cm, and the *x*-axis is the TVDI value extracted from the location of the sampling point. It can be seen from Figure 3 and Figure 4 that there is a clear negative correlation between soil moisture and TVDI in the summer from 2010 to 2015. With increased soil moisture, TVDI shows an obvious negative trend. We found TVDI to have a significant linear relationship (*p* < 0.01) with soil moisture in each year. Moreover, in comparing Figure 3 and Figure 4, we found that the correlation between TVDI data retrieved by geomorphological division and soil moisture data was higher than that between TVDI data retrieved directly without geomorphological division and soil moisture data, and the R^2^ value of the former were all over 0.41. Therefore, the TVDI data retrieved by the geomorphological division method can better reflect the soil moisture status, and also show that the TVDI method is suitable for studying the drought status of the QTP.

## 4. Results

### 4.1. Characteristic of NDVI-LST Feature Space

Figure 5 shows the feature spaces of the eight geomorphological divisions that we constructed, which basically satisfied the triangular or trapezoidal space theory. The maximum and minimum LST values corresponding to NDVI were extracted through the TVDI VTCL plug-in of the ENVI 5.3 software, and the NDVI-LST space was constructed. When constructing the NDVI-LST feature space of the QP, the range of NDVI value from 0 to 1 was selected. It should be noted that when constructing the NDVI-LST feature space of the other seven regions, we excluded values of NDVI < 0.2 from the analysis. After determining the NDVI range of each district, we set the sample size in the plug-in to 1 to obtain the annual and quarterly NDVI-LST feature spaces from 2000 to 2019. Because the NDVI-LST feature space of the eight geomorphological divisions in 2001 had the best fit (the average R^2^ is 0.69), we took the characteristic space of 2001 as an example. Figure 4 shows that there was an obvious linear relationship between the maximum and minimum surface temperature and NDVI. With increased NDVI, the maximum surface temperature decreased gradually, while the minimum surface temperature increased slowly. It can be seen from Appendix B Table A3 that the average R^2^ of the HDM was 0.35, with the low value of R^2^, poor relative fitting, and low accuracy. Other regions have high R^2^ values, a good fit, and high accuracy, among which the QHB, QP, and HMLY were the best (the average value of R^2^ is 0.75).

### 4.2. Temporal and Spatial Variation Characteristics of TVDI

The dry-edge and wet-edge equations were used to calculate the TDVI value of each pixel at different times, and the TVDI value was applied as the classification index of drought in the QTP [58]. TVDI was divided into five grades (Table 4) [21,24,59]: extremely wet (0 ≤ TVDI ≤ 0.2), wet (0.2 < TVDI ≤ 0.4), normal (0.4 < TVDI ≤ 0.6), dry (0.6 < TVDI ≤ 0.8), and extremely dry (0.8 < TVDI ≤ 1.0). When the TVDI was between [0, 0.6], the drought type of the region was no drought; when the TVDI was between (0.6, 0.8], the drought type of the region was drought; and when the range was between (0.8, 1.0], the drought type of the region was severe drought.

#### 4.2.1. Spatial Variation Characteristics of TVDI

Figure 6 shows that there were obvious regional differences in the spatial distribution of the mean value of TVDI in the QTP from 2000 to 2019, which was mainly characterized by drought in the southwest and north, and wet conditions in the northeast and southeast. Overall, the average annual value of TVDI in the QTP was 0.56, which was normal. However, the regional drought situation was not consistent. For example, for QHB the average annual TVDI was 0.66, which we classified as being in drought. The no-drought areas were distributed in the eastern region of the STR, HMLY, HDM, the Qilian Mountains, and the Kunlun Mountains. Most of these areas were forest and grassland, accounting for 56.9% of the area of the QTP. The drought areas were mainly distributed in the southwest QP, in the west HMLY and most of the KWKM, and grassland was the main land use type, accounting for 33.4% of the area of the QTP. The severe drought areas accounted for 9.7% of the QTP, and were mainly distributed in the Qaidam Basin, the western part of AQM, and the river valleys of HMLY.

The QTP was drier in spring and summer and humid in autumn and winter (Figure 7). Spring was the driest season of the year. The southern part of the QTP was humid, and the southwestern and northern parts were dry. Most of the QHB, the southern part of the QP, and the central and western parts of the HMLY were relatively dry. In summer, the QTP was generally wet in the south and dry in the north, in which drought severity in the southwest was reduced and drought severity in the north increased. In terms of spatial distribution, the QHB, the Altun Mountain, the west of HMLY, and the southwest of QP were relatively arid. In autumn, extremely wet areas in the QTP decreased, while wet and normal areas increased. Dry areas were concentrated in the north and southwest, such as the QHB, QP, and the west of HMLY. In winter, the central part of the QTP was wet, the western part was dry, and the wet areas increased. Winter was the wettest season of the year with the smallest arid area. Only the south of QP and the mid of HMLY were in a dry state. In short, the AQM and QP were prone to spring drought, the HMLY was prone to autumn drought, the QHB was prone to spring drought and summer drought, and the central QHB was in a state of drought for a long time, and other areas had no obvious seasonal drought.

#### 4.2.2. Temporal Variation Characteristics of TVDI

The QTP has exhibited an obvious trend in drought severity over the past 20 years, and the central region has become the center of increased drought (Figure 8). The area with aggravated drought accounted for 52.9% of the area of the QTP, the area with a significant increase in TVDI accounted for 5.8% of the QTP, and the area with a significant and extremely substantial increase accounted for 2.3% of the QTP. These two types of drought variation are mainly distributed in the southeast of QP, the southwest of STR, the southwest of AQM, the most of KWKM, and the south of CEKM.

The land use types in these areas were mainly unused land and grassland, and the regional characteristics were high elevation and low vegetation cover. The areas with non-significant increase in TVDI accounted for 44.8% of the total area, mainly distributed in the KWKM, CEKM, QP, STR, and other areas. The areas with non-significant decrease in TVDI were mainly distributed in the northwest of HMLY, HDM, and QP. Most of these areas were grassland and forest, and accounted for 47.1% of the whole area. In general, there were obvious differences in drought severity trends in the QTP, with significant drying trends in the eastern and southeastern parts of QP and the western and southwestern parts of STR. There was a risk of drought in the Kunlun Mountains and its north, and the risk of drought in the HDM and HMLY regions decreased.

The QTP had a drying trend in all seasons except autumn (Figure 9). Spring was the season with the largest drying trend. The AQM, CEKM, STR, and HMLY were the centers of increasing drought in spring. The land use types in these areas are mainly grassland and unused land, accounting for 61.7% of the total area. The areas with a drying trend in summer were mainly distributed in the mid of HMLY, the southeast of QP, and the southwest of STR, which accounted for 60% of the entire region. Autumn was the season with the smallest area with a drying trend, which accounted for 48.9% of the total area of the region without a drying trend. There were three areas with strong drying trends in winter, QP, QHB, and AQM, and the areas with drying trends accounted for 58.2% of the total area. Overall, there was a significant drying trend in spring, the drought risk in the northern QTP weakened in summer, but the drought risk in the southern part remained very high. In autumn, most of the areas were the wettest and had a weaker trend in drought. In winter, the risk of drought in the northern QTP increased significantly.

#### 4.2.3. Spatial and Temporal Variation Characteristics of TVDI Based on Cluster Analysis

According to the distribution and variation characteristics of TVDI in various regions of the QTP, the method of cluster analysis was used to deeply explore the law of TVDI variation in the QTP. Based on annual average TVDI, TVDI_Slope of the annual average TVDI, and the trend classification of the annual average TVDI, we selected nine indicators in each district, maximum and minimum annual average TVDI (TVDI-max and TVDI-min), mean annual average TVDI (TVDI-mean), the standard deviation of annual mean TVDI (TVDI-SD), the mean value of TVDI_Slope (slope-mean), the area proportion of areas with non-significant decrease (TVDI-t3), the area proportion of areas with non-significant increase (TVDI-t4), the area proportion of significantly increased areas (TVDI-t5), and the area proportion of areas with extremely significant increase (TVDI-t6) (Table 5). We then used the Pearson correlation to conduct a case-by-case systematic cluster analysis (Figure 9).

The TVDI of the QTP demonstrated obvious gradients in its characteristics. It can be found from Figure 10 that the QTP are divided into three categories: HMLY and HDM are one category, QP, STR, and AQM are one category, and the last category includes KWKM, CEKM, and QHB. The HMLY and HDM areas are located in the southern part of the QTP, and their annual average TVDI values were 0.59 and 0.52, respectively, and the drought severity was at the non-drought level. The standard deviation of their annual TVDI was not much different, the mean value of slope was less than 0, and the areas with non-significant decrease had the largest ratio of area to the total area. They were characterized by a low degree of drought and a trend towards wetter conditions.

The QP and STR are in the central QTP, and the AQM is in the northern part of the QTP. The average TVDI of the three areas was between 0.5 and 0.6, and the drought category was normal. The average slope was large, and the proportion of the area with non-significant decrease to the total area was between 41.10% and 47.34%. These three areas were characterized by a lower degree of drought and increased regional drought risk.

The KWKM, CEKM, and QHB are in the northern-central QTP. The average value of QHB was relatively high, and the drought severity was drought. The standard deviation of the annual average TVDI of the three places ranged from 0.18 to 0.21, and the average value of slope was between 1.04 and 1.54. Among the three regions, the difference in results was small. The ratio of the non-significant decrease area to the total area decreased in the three regions, and the ratio of the non-significant increase area to the total area was the highest, which indicated that the three regions were characterized by a higher degree of drought and had a significant risk of drought in the region. In summary, the drought severity and drought risk in the QTP increased from the southeast to the northwest, which indicated that the spatial variation pattern of TVDI was roughly consistent with the spatial variation of precipitation and temperature in the southeast–northwest gradient.

### 4.3. Characteristics of TVDI Drivers

#### 4.3.1. Climate Driven Characteristics of TVDI

Figure 11 shows that the temperature, precipitation, and TVDI of the QTP had different trends. Annual precipitation had a negative trend, average annual temperature had a positive trend, and the average annual TVDI had a positive trend. The changes of these three indicated that the study area had a tendency of drought. From 2000 to 2017, the annual precipitation in the QTP had a negative trend, with a trend of −0.39 mm/year. The annual precipitation in the QTP decreased from the southeast to the northwest. Among them, the HMLY and HDM had the most precipitation, and the Qaidam Basin, AQM, and the western QP had the least precipitation. From 2000 to 2017, the average annual temperature of the QTP has fluctuated upward at a rate of 0.02 °C/year. The spatial distribution of the average annual temperature for many years was uneven. The areas with lower temperature were distributed in the Kunlun Mountains, the northern QP, and STR, and the areas with higher temperature are distributed in HMLY, the Qaidam Basin, and HDM. From 2000 to 2019, the TVDI had an insignificant upward trend, with a cyclical change of roughly six years.

There are regional differences in the effects of precipitation and temperature on the TVDI of the QTP. The influence of meteorological factors on TVDI was quantitatively analyzed using the partial correlation coefficient of TVDI, annual precipitation, and annual average temperature obtained by pixel-by-pixel calculation. Figure 12a shows that the partial correlation coefficients between TVDI and annual precipitation in the QTP ranged from −0.97 to 0.93. The areas where TVDI were positively correlated with annual precipitation accounted for 53.56% of the total area of the whole region, which were distributed in the HMLY, STR, and QHB. The areas with a negative correlation accounted for 46.44% of the total area, which were mostly distributed in the QP and HDM. According to the spatial characteristics of partial correlation between TVDI and average annual temperature in the QTP, Figure 12b shows that the partial correlation coefficient between TVDI and temperature ranged from −0.89 to 0.98. Among them, the areas with a positive correlation accounted for 64.06% of the entire region, and most of them were distributed in the QP, STR, and the east wing of the HMLY, KWKM and CEKM. The areas with a negative correlation accounted for 35.94% of the area and were mainly distributed in the west wing of the HMLY and HDM.

After calculating the *t*-test of the partial correlation coefficient and the F-test of the multiple correlation coefficient, according to Table 1, the driving force of the climatic factor of the QTP was obtained (Figure 13). The factors affecting the change in drought severity in the QTP were complex. The drought in most areas was affected by other factors. The areas driven by a single factor of temperature or precipitation were distributed in clusters, and the areas dominated by temperature and precipitation were scattered. The distribution of the areas driven by precipitation was concentrated, accounted for 1.32% of the total area, and were mainly located in the Qaidam Basin and scattered in the STR and HMLY. The land use types in these areas were mainly unused land and the vegetation coverage was low. The areas driven by temperature accounted for 3.65% of the total area of the region, which were distributed in all regions of the QTP and concentrated in areas with higher elevation, such as QP. The areas driven by precipitation and temperature account for 5.95% of the total area, which were widely distributed in various regions of the QTP, mostly in HMLY, HDM, and AQM.

#### 4.3.2. Driver Characteristics of TVDI

Figure 14 shows that the evolution of drought in the QTP was affected by various factors, but different topography, vegetation types, and other factors led to differences in the main driving factors of each geomorphological division. The drought situation in the AQM was mainly driven by precipitation, and secondly affected by temperature. In the KWKM area, the driving effect of the Euclidean distance factor to the water surface body was more significant. The driving factors of QP drought were mainly temperature and Euclidean distance of water surface body, and other factors had less influence. The drought severity in the HMLY was driven by precipitation, followed by temperature and elevation. The drought conditions of QHB, CEKM, STR, and HDM were comprehensively affected by precipitation and temperature, and other factors had relatively little effect. In general, the influence of climatic factors (temperature and precipitation) on the drought conditions of the QTP was relatively stable, geomorphological factors (slope, aspect and elevation) played an important role in the drought conditions in each region, and accessibility factors (Euclidean distance of water surface body and Euclidean distance of built-up land) and land use factors had relatively little influence.

According to the random forest model, we obtained a local dependence map of the driving factors in each geomorphological division (Figure 15), to further explore the relationship between each driving factor and TVDI. We found that the overall trends of the relative importance of driving factors in each region were similar, but there were obvious differences in the sensitivity intervals of the driving factors.

First, the relative importance of annual average temperature increased between −20 °C and 20 °C, and maintained a high level from 0 °C to 16 °C. The relative importance of HDM peaked at 16 °C, and the relative importance of other regions peaked from 0 °C to 5 °C.

Second, the relative importance of annual precipitation fluctuated and decreased from 0 mm to 1500 mm, and the impact on regional drought was more profound when the precipitation was less than 500 mm. The trend of the relative importance of the QHB and CEKM showed a double peak, and the trend of the relative importance of other regions had a single peak.

Third, the relative importance of elevation had a fluctuating and negative trend overall, but there were obvious differences in the sensitive range of elevation in each area. The relative importance of HMLY elevation had two peaks, about 0 m and 4700 m, respectively, and the relative importance of elevation in other areas ranged from 3000 m to 5200 m.

Fourth, the relative importance of the slope changed similarly in different areas, with an upward trend from 0° to 5°, a downward trend from 5° to 30°, and remained stable above 20°.

Fifth, the relative importance of the aspect had a clear peak, rising first and then falling between 0° and 360°, and had a high level from 175° to 200° (sunny slope).

Sixth, the relative importance of the Euclidean distance of built-up land did not change significantly in the HMLY, KWKM, and QHB, and the relative importance of other regions had a downward trend. The Euclidean distance of built-up land was the most sensitive from 0 m to 1000 m. Seventh, the relative importance of the Euclidean distance of water surface body was on the rise, at a high level from 1600 m to 3000 m, and remained stable above 3000 m. Finally, the relative importance of land use types varied significantly among different types, with grassland and cultivated land generally being the highest, built-up land and other land use being higher, and the forest and water surface body being the lowest.

## 5. Discussion

### 5.1. Analysis of NDVI-LST Feature Space

Considering the characteristics of LST and NDVI data in the study area, determining a reasonable range of NDVI for regression fitting and obtaining the appropriate dry- and wet-edge equation was the key for calculating TVDI. When the vegetation coverage is less than 15%, NDVI is a poor indicator of vegetation cover in the region [60]. Moreover, when determining the NDVI range in the feature space, the corresponding LST minimum value in the low-value area of NDVI has a tail decline phenomenon [61]. The above two phenomena may lead to the deviation of the simulated dry and wet edge from the theoretical boundary. Therefore, when determining the NDVI range of the feature space in our study, to avoid regional differences and the occurrence of tail subsidence, except for the QP, we did not use NDVI values < 0.2 in the other seven regions, which ensured that the dry and wet edges of each region conformed to the theoretical boundary.

Combined with Figure 4 and Appendix B Table A3, it can be concluded that the feature spaces in different phases and different regions were triangular. Thus, as NDVI increased, the maximum and minimum values of LST gradually approached and converged at one point. The overall characteristics of the data in the study area were consistent, so the NDVI-LST relationship could be used to accurately simulate the drought state [62]. The fitting of the dry and wet edges in the eight geomorphological regions were all good, which indicated that the NDVI-LST feature space of each region can accurately reflect the LST distribution and NDVI characteristics. Among them, the dry- and wet-edge coefficients (R^2^) of the QHB, QP, and HMLY were the most stable and had the best fit, while the dry- and wet-edge coefficients (R^2^) of the HDM fluctuated greatly and the fit was relatively poor, which may be related to deep-cut landforms and higher vegetation coverage [23].

### 5.2. Analysis of Spatial and Temporal Variation Trend of TVDI

The overall spatial distribution of the QTP was drought in the southwest and north, and wet in the northeast and south, which was consistent with previous research results [31]. The QTP was drier in spring and summer, and wetter in autumn and winter. In spring and summer, the western part of the QTP was relatively dry, which was caused by the vegetation in the growing season, a large demand for water, high temperature, high evapotranspiration, and less precipitation in the region. In autumn, with the decrease of temperature and water evaporation, the water required for vegetation growth decreases, and snowfall occurs in high-elevation areas, which replenishes water resources, increasing the soil water content in the central part of the QTP, thus alleviating the drought. In winter, under the combined influence of land-source water vapor and ocean-source water vapor, the central region of the QTP is obviously humid [63]. In addition, it is also affected by factors such as the decrease of temperature, vegetation dormancy, the expansion of snow cover area and the increase of snow depth [64].

In the past 20 years, the degree of drought in more than half of the QTP has increased significantly, and the central QTP has become the center of aggravated drought. Other studies have demonstrated similar findings [65,66]. The decrease in precipitation and increase in evapotranspiration in the QTP decreased soil moisture and caused a drying trend [67]. The QTP demonstrates a drying trend in other seasons except autumn, which was also concluded by Feng et al. [35] using the SPEI method. As the seasons changed, the drought risk varied in different regions, such as AQM. The AQM was prone to spring drought and combined with the drying trend in the region, the risk of drought was further increased.

Affected by the climatic gradient, the drought pattern of the QTP generally had a southeast–northwest gradient. According to the statistical characteristics of TVDI in eight geomorphological areas, the QTP could be divided into three categories by cluster analysis. The first category included the HMLY and HDM, which had a low degree of drought and a wetting trend. The second category included the QP, STR, and AQM, which had a low degree of drought and a drying trend. The third category included the KWKM, CEKM, and QHB, which had a higher degree of drought and an increased risk of drought. The drought severity and variation characteristics of these three regions were different, which was not only related to differences in precipitation, temperature, elevation, land use type, and vegetation cover, but also affected by significant geomorphological effects. As the Yangtze River, Yellow River, Yarlung Zangbo River, and other rivers flow through the HMLY and HDM areas, there was abundant precipitation, sufficient water resources and high soil humidity. The QP, STR, and AQM are located in inland areas, with high elevation, low temperature, and less precipitation. However, lakes, glaciers, and other water bodies have large areas, so water resources are more abundant and the regional drought is low. The KWKM, CEKM, and QHB are located deep inland and receive little precipitation. The land use type is dominated by grassland and unused land and drought conditions are susceptible to climate change.

### 5.3. Analysis of Driving Force of TVDI Change

The change of TVDI in the QTP was mainly affected by climate factors, but there were slight differences in the driving factors of TVDI in each geomorphological division. The AQM and HMLY were dominated by precipitation, the QP was dominated by temperature, and the QHB, CEKM, STR, and HDM were affected by temperature and precipitation. The driving factor of TVDI in the KWKM was the Euclidean distance of water surface body, which was due to the wide distribution of glaciers and lakes affected by monsoon [68]. The leading factors of TVDI in all regions of the QTP echo the results of climate drivers by partial correlation analysis, which also indirectly showed the reliability of the random forest method. In addition, the random forest results of TVDI data obtained by direct inversion without geomorphological division also showed that the QTP was dominated by climate factors, followed by geomorphological characteristics, which was consistent with the results we obtained.

#### 5.3.1. Meteorological Factors

There were various factors affecting drought in the QTP, among which precipitation and temperature played a significant role [69,70]. Precipitation affects water resources, and temperature affects evapotranspiration, which interact and jointly affect drought in the QTP. In recent years, the decrease of precipitation and the increase of temperature in the QTP have changed the soil moisture [71], which has affected the growth of vegetation, weakened the ability of soil to retain water, and led to increased drought in the QTP. Among them, increased temperature has increased snow melt and evapotranspiration in the study area [41], which is an important reason for the spatial-temporal change of drought in the QTP [72]. Combined with analysis of relative importance, there were large spatial differences in the relationship between TVDI and climate factors in the QTP. When the annual average temperature was greater than 0 °C, the relationship between the TVDI and the annual average temperature of the QTP was the most sensitive. When the annual precipitation was less than 500 m, this factor had the most important impact on the change of TVDI. The areas with high temperature and low precipitation are in the western parts of HMLY, the southern parts of QP, and the northwestern parts of KWKM and the QHB. Due to the combined influence of precipitation and temperature, these regions are prone to drought and extreme drought.

Climate change largely affects the duration of drought [73]. A warming climate will increase the risk of drought [74], especially the frequency and duration of drought in arid areas [75]. The reduction of regional precipitation will lead to stronger and more frequent drying periods and the intensification of evaporation induced by global warming, and this situation will increase the probability of drought in the region [76]. From the climate driven zone map, it was found that drought in the central QTP was dominated by temperature, and drought in the northern basin was driven by precipitation, of which temperature was more driven than precipitation. The Qaidam Basin is located deep within the interior of the continent and is surrounded by high mountains, which affects the moist air flow into the basin, resulting in less precipitation in the region. The TVDI in QHB was dominated by precipitation and had a positive correlation with the annual precipitation. With decreased precipitation, the drought in this area increased. The drought in QP was dominated by temperature, and there was a positive correlation between TVDI and temperature in this area. Affected by the temperature rise of the QTP in the recent 20 years, QP has become the central area with an enhanced drying trend.

#### 5.3.2. Geomorphological Factors

Geomorphological characteristics are important factors that affect the construction of regional basic factors such as mountains, climate, vegetation, roads, and human activities. The uplift of the QTP has not only formed a unique plateau climate [77], but also had a profound impact on the composition of modern atmospheric circulation patterns, the establishment of the Asian monsoon system [78], regional and global climate change, the development and evolution of desert loess, plateau ice and snow, lake and river water systems and ecosystems, and the origin and evolution of human beings [79]. Among them, the thermal effect of the QTP has an impact on the Asian monsoon and precipitation variability [80], resulting in regional differentiation of precipitation and heat generation [81], which directly or indirectly cause drought, water shortage, and deserts. The distribution of water and heat in the region affected by elevation results in a gradient of drought conditions in the QTP [82], especially in HMLY. The HMLY Valley area is blocked by the southern high mountains to the water vapor from the North Indian Ocean, resulting in a large amount of precipitation falling on the windward slope, and the area has been in a dry state for a long time. At elevations between 3000 m and 5200 m, the impact on TVDI is most significant, with rainfall and temperature changing positively in the high-altitude range [35].

Slope affects soil properties, soil nutrient content, and vegetation types to a certain extent. With the increase of slope, precipitation infiltration is difficult, and the loss rate of soil water intensifies, which further affects the likelihood of drought in the region [83]. The smaller the slope, the better the soil moisture retention. Slopes less than 5° were the most sensitive to TVDI in the study area. Such areas are mainly distributed in the central and northern QTP, with high elevation and local precipitation less than 500 mm, and slope had a low degree of influence on TVDI.

Aspect affects the amount of solar radiation, water content, and evapotranspiration received by the slope, resulting in aspect-related differences in drought conditions [40]. When located on a sunny slope, TVDI was most sensitive to aspect. Compared with the shady slope, the sunny slopes have longer sunshine hours, receive more solar radiation, and better develop vegetation, which increases the regional evapotranspiration intensity, reduces soil water content, and causes drought differentiation. Compared with factors such as temperature, precipitation, and elevation, the aspect had less influence on the TVDI, which indicated that the humidity change caused by the aspect had a relatively small impact on the regional drought change.

#### 5.3.3. Accessibility Factors

With economic development and the expansion of urban land, the impact of human activities on regional drought has increased [84,85,86,87] and the impact of human activities on drought is greater than the duration of drought [88]. There are two aspects to the impact of human activities on drought. First, human activities directly act on surface water and groundwater resources, consume and change the occurrence and regeneration conditions of water resources, and cause or exacerbate drought [89]. Second, due to urban and rural construction, traffic route development, deforestation, wasteland removal, and other activities that change the land surface, the soil moisture and roughness are reduced and the surface reflectivity of the underlying surface is increased, thereby changing the production and confluence law and evaporation composition, which ultimately leads to localized reductions in precipitation and increased drought. In terms of the accessibility impact of cities and towns, the closer to cities and towns, the more intense human activities become and the greater their effects on landform and surface cover. Therefore, the impact of the Euclidean distance of built-up land demonstrated significant distance attenuation, and its sensitive range was mostly between 0 m to 1000 m. Due to the relatively small proportion of built-up land in the QTP, it had little impact on the overall changes in drought on the QTP.

There are many frozen soils, glaciers, and lakes in the QTP, especially in KWKM and QP. Soil moisture in the areas surrounding rivers and water bodies gradually declines as one moves further away from the water bodies, which causes a wet to arid trend as the distance from water bodies increases. Under the background of global warming and the imbalance of water towers in Asia, the glaciers in the QTP are in a state of continuous melting, their solid water is melting rapidly, and their liquid water is increasing [90]. This affects the hydrological process in the study area and relieves the drought pressure in some areas [91]. In our study, the Euclidean distance to water surface body was used to analyze the impact of glaciers, lakes, and other waters on TVDI. We found that the Euclidean distance of water surface body had the characteristics of distance attenuation on the impact of regional drought. When it was greater than 1600 m, the relationship between TVDI and this factor was the most sensitive. In addition, the formation and melting of glaciers and permafrost produce obvious seasonal differences in the drought situation of the QTP. In summer, the QTP had high temperature, high evaporation, and a large demand for water by vegetation, but its drought degree was less affected by glacier and frozen soil melting than in spring [92]. In winter, the soil of the QTP gradually freezes, the regional evapotranspiration decreases, and the soil water holding capacity increases, making winter the wettest season of the year. In the short term, glacier melting will alleviate drought in some areas, but long-term glacier melting will reduce the ability of arid areas to cope with drought and increase the risk of drought.

#### 5.3.4. Land Use Type Factor

Differences in vegetation cover and surface roughness of different land use types lead to differences in their impact on TVDI [93]. The results of local dependence analysis showed that grassland and cultivated land had the greatest impact on TVDI, followed by urban and rural built-up land and other land, and that the forest and water bodies had the lowest impact. The crops in the QTP are mainly highland barley, and the sowing and growing season spans from March to September. The cultivated land is generally adjacent to built-up land and is greatly affected by human activities [94]. The ecosystem stability of cultivated land and built-up land is relatively weak, and the soil water holding capacity is weak, which affects the change of TVDI [95]. Forest is mainly distributed in the southeastern QTP, which has rich water resources, good soil coverage, less evaporation, strong soil water holding capacity, and non-significant impact on TVDI [96]. The water area includes lakes, permanent glaciers, reservoirs, ponds, beaches, and other water bodies, which are humid and have low drought risk. Since the water changes to the surrounding area, the soil moisture gradually decreases with a trend from wet to drought. Grassland and unused land are the most widely distributed in the study area and are easily affected by climate fluctuations, have relatively low vegetation coverage, poor stability, and strong ground evaporation, and thus have a more significant impact on TVDI.

### 5.4. Applications and Limitations

As an important ecological security barrier in China, the QTP plays a role in water conservation, biodiversity protection, and soil and water conservation. It is a ‘sensor’ of climate change in Asia and the Northern Hemisphere. Drought affects regional vegetation status, the water cycle, and human life, and can thereby destroy the stability of the ecological environment and restrict social and economic development. Using geomorphological division, we applied the TVDI method to identify the temporal and spatial variation of drought in the QTP from 2000 to 2019 in an effort to provide a scientific basis for the prevention and control of regional drought and promote sustainable development of the region. Under the background of global warming, we found that the QTP had a tendency of drought, and the central QTP has become the hotspot of increased drought severity. In response to the potential challenges of drought risk, the government should actively adjust and optimize drought prevention and control measures. In areas with low drought risk, such as HDM and HMLY, the government should strengthen the ecological protection of cultivated land and important areas such as rivers, lakes, and surrounding areas, control the intensity of human reclamation activities, and closely monitor changes in the hydrological conditions of grasses. In areas with perennial droughts, such as QHB, KWKM, and CEKM, where the drying trend was aggravated, a multi-geographical synchronous monitoring system should be established to identify and monitor drought dynamics. In sandy and bare areas, ecosystems need to be restored. In agricultural farming areas, the planting structure needs to be optimized, and high-efficiency, water-saving agricultural methods need to be developed. In pastoral areas, the scale and quantity of grazing should be controlled, and the protection of grasslands should be strengthened by taking measures such as zoning and rotation grazing.

There were some aspects of our study that could be improved. In terms of the TVDI calculation, the TVDI of each area was calculated based on the geomorphological division, and then the TVDI of the whole area was obtained by mosaic. However, this method led to discontinuity of the TVDI value at the geomorphological edge, or the edge effect. Although we used the TVDI classification to weaken the edge effect, the moving window method can be tested to eliminate the edge effect in the future. In terms of index selection, the effects of natural factors, such as stratigraphic lithology, geological structure, surface water, and groundwater have not been fully considered, and the impact on human activities has not been adequately described. Future research can further explore the mechanisms of drought in the QTP and divide the regional drought risks based on drought severity and drying trends. Methods such as GWR can be used in comparison with or combined with RF methods to improve the accuracy of model interpretation.

## 6. Conclusions

Here, we selected the QTP as the study area. However, considering the influence of the huge spatial scale effects of the study area on TVDI inversion, we divided the QTP into eight geomorphological areas. TVDI inversion was carried out for each area and the data for the whole area were synthesized using the TVDI grading method. Partial correlation and multiple correlation analyses were used to explore the effects of precipitation and temperature on TVDI. Then, eight factors, including meteorology, geomorphology, land cover, and human activities, were selected to construct a random forest model to identify the mechanisms driving changes in TVDI. We gathered four main findings.

First, in the past 20 years drought frequency and severity in the QTP has increased, especially in the central region. The spatial distribution of TVDI was wet in the southeast and arid in the northwest, with conspicuous regional differences. The main areas with increasing TVDI were distributed in the southeast of QP, the southwest of STR, and the Kunlun Mountains. The main areas where TVDI remained unchanged and decreased were distributed in the HDM and HMLY where vegetation cover was high.

Second, the seasonal differences of drought were significant in the QTP. Spring was the driest season of the year, and there was a significant drying trend in the central and northern QTP. It was relatively dry in summer, the drought risk in the north of QTP was weakened, and the risk in the south was still very high. Autumn was relatively wet, and the region had a downward trend in drought severity. Winter was the wettest season of the year, but the risk of drought increased significantly in the northern part of the QTP.

Third, the spatial pattern of TVDI changes in the QTP was generally consistent with changes in precipitation and temperature, which had a gradual change from southeast to northwest. According to the spatial distribution and variation of aridity, we divided the study area into three categories. The first category, the HMLY and HDM in the southern region, had low drought severity and was becoming more humid. The second category, the QP, STR, and AQM in the central and northern regions, had low drought severity but had a positive trend in drought severity. The third category, the KWKM, CEKM and QHB in the north-central region, had higher drought severity and a positive trend in drought severity.

Finally, the driving effect of each factor on TVDI was significantly different in the QTP. Except in the KWKM, temperature and precipitation were the dominant factors in the variation of TVDI in the QTP, and regions with higher temperature and less precipitation had a more significant impact on TVDI. Geomorphological factors play an important role in the change of TVDI. The areas with an elevation of 3000~5200 m, slopes less than 5°, and sun-facing slopes had a more significant impact on the change of TVDI. The influence of the Euclidean distance of urban and rural built-up land and water bodies showed distance attenuation, and the sensitive areas of these two factors were <1000 m and >1600 m, respectively. Among the land use types, grassland and cultivated land had a more significant impact on TVDI changes, while forest and water bodies had less impact on TVDI changes.

## Figures and Tables

**Figure 1 ijerph-19-07909-f001:**
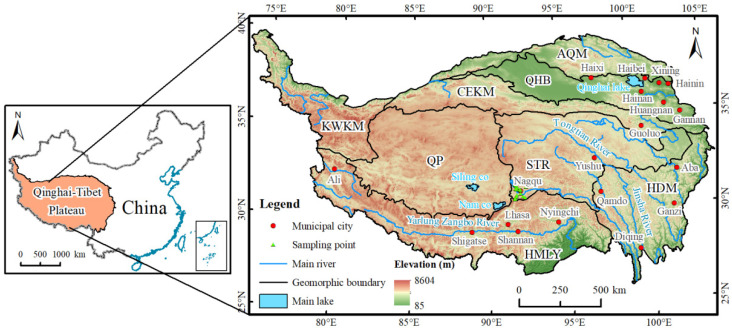
Map of the Qinghai–Tibet Plateau (QTP). AQM is the high valley area of the Altun–Qilian Mountains. QHB are the high mountain areas of Qaidam-Yellow River-Huangshui River Basin. KWKM are the high mountain areas of Karakorum and Western Kunlun Mountains. CEKM are the high mountain areas of the central and eastern Kunlun Mountains. QP are the lake and basin areas of the Qiangtang Plateau. STR are the mountains sources of the Yangtze River, Yellow River, and Lancang River (Three Rivers or Sanjiangyuan) and the valley bottom of the upper reaches of Three Rivers. HMLY are the high mountain areas of the Himalayan Mountains. HDM are the high mountain and valley areas of Hengduan Mountains. (Appendix A of Table A1 for acronyms).

**Figure 2 ijerph-19-07909-f002:**
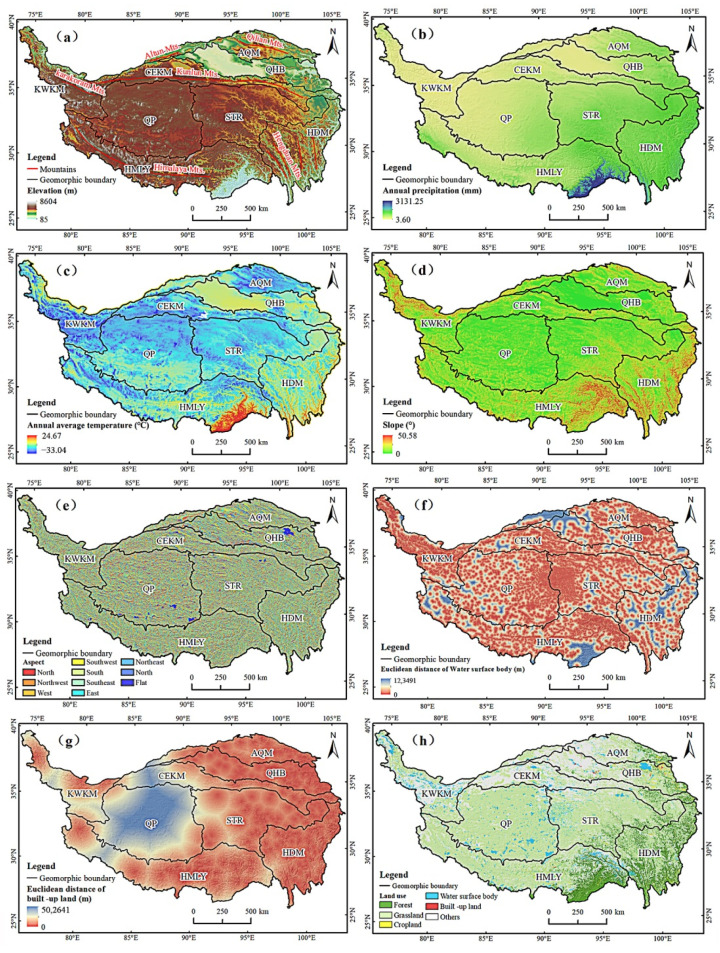
Maps of the eight factors. (**a**) Elevation; (**b**) Annual precipitation; (**c**) Annual average temperature; (**d**) Slope; (**e**) Aspect; (**f**) Euclidean distance of water surface body; (**g**) Euclidean distance of built-up land; (**h**) Land use and Land cover.

**Figure 3 ijerph-19-07909-f003:**
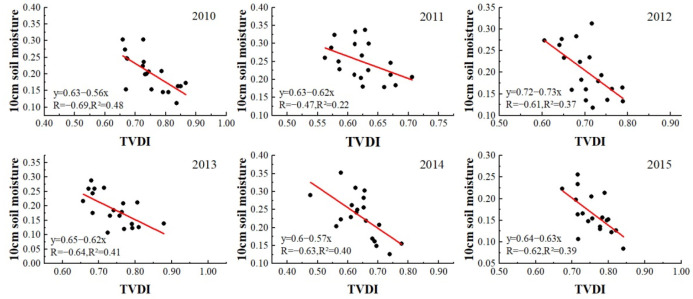
Correlation between TVDI retrieved directly without geomorphological division and 10 cm soil moisture in summer from 2010 to 2015.

**Figure 4 ijerph-19-07909-f004:**
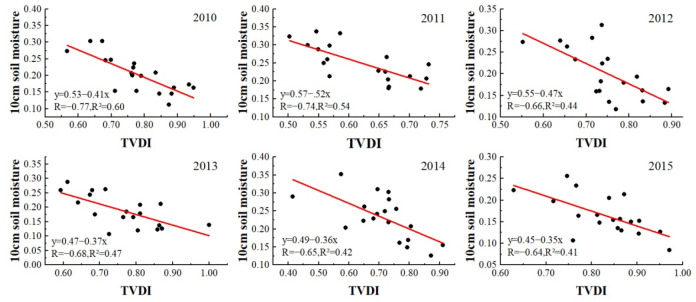
Correlation between TVDI retrieved by geomorphological division and 10 cm soil moisture in summer from 2010 to 2015.

**Figure 5 ijerph-19-07909-f005:**
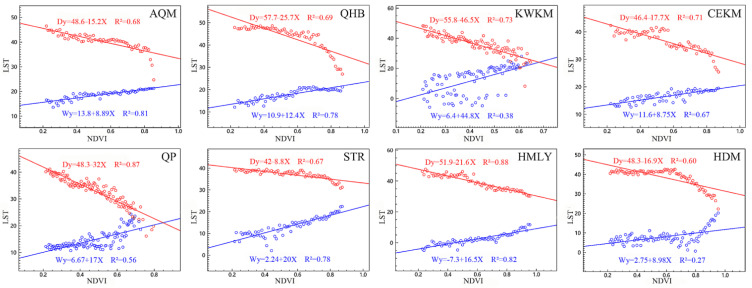
NDVI-LST spatial characteristics of eight geomorphological divisions of the QTP (taking 2001 as an example)(The horizontal axis represents NDVI. The vertical axis represents LST. The blue line is the fitted wet edge equation. The red line is the fitted dry edge equation. The sample points are represented by small hollow circles).

**Figure 6 ijerph-19-07909-f006:**
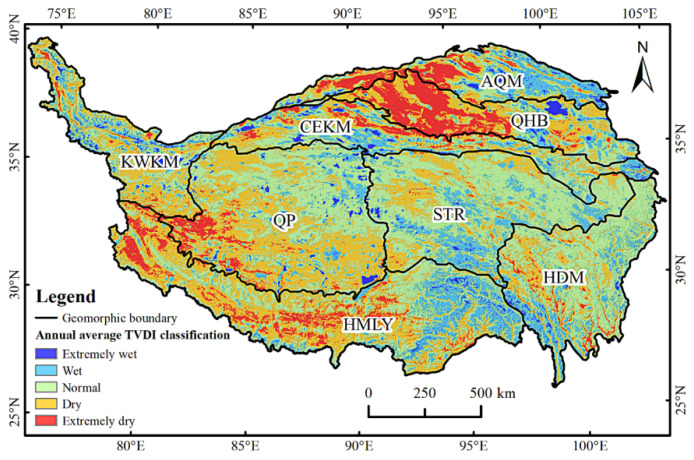
Annual average TVDI classification of the QTP.

**Figure 7 ijerph-19-07909-f007:**
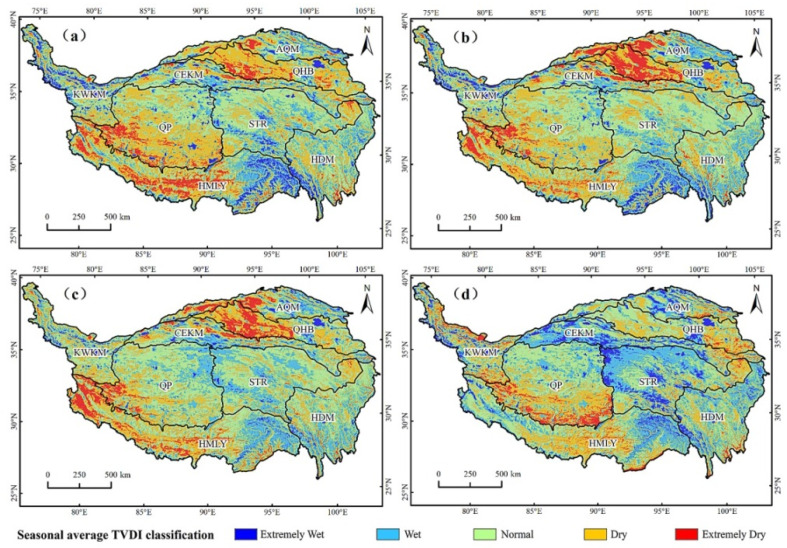
Classification of seasonal average TVDI. (**a**) Spring; (**b**) summer; (**c**) autumn; (**d**) winter.

**Figure 8 ijerph-19-07909-f008:**
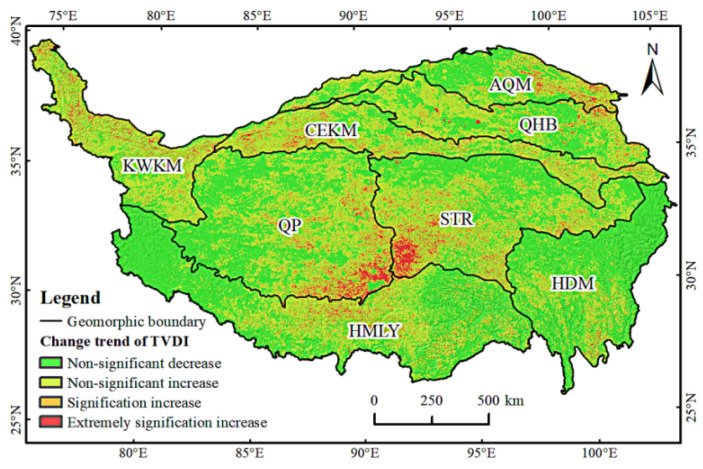
TVDI trends in QTP.

**Figure 9 ijerph-19-07909-f009:**
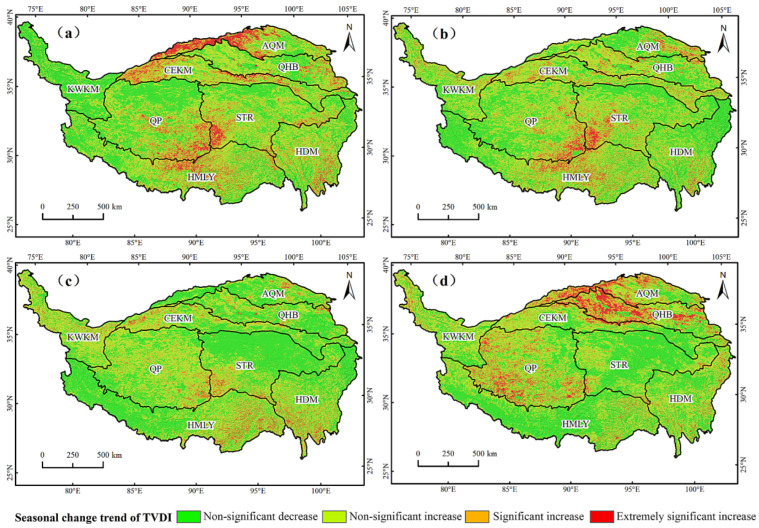
Seasonal TVDI trends of the QTP. (**a**) Spring; (**b**) summer; (**c**) autumn; (**d**) winter.

**Figure 10 ijerph-19-07909-f010:**
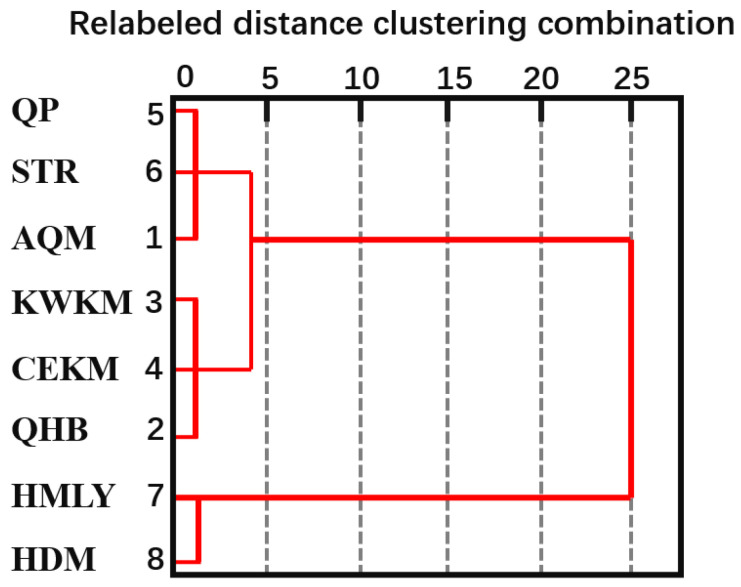
Pedigree of systematic cluster analysis.

**Figure 11 ijerph-19-07909-f011:**
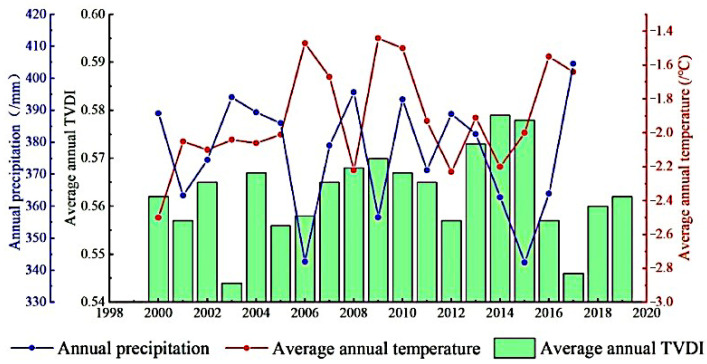
Changes in annual precipitation, average annual temperature and average annual TVDI of the QTP.

**Figure 12 ijerph-19-07909-f012:**
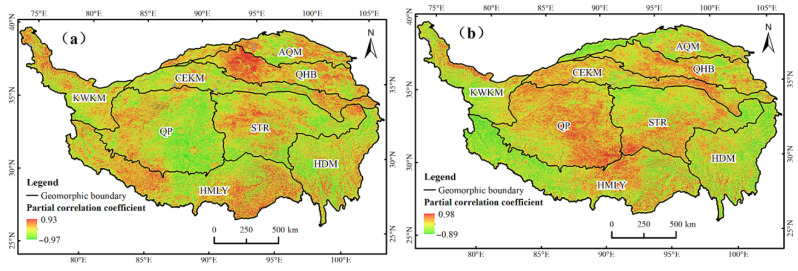
The distribution of partial correlation coefficients between TVDI and annual precipitation (**a**) and annual average temperature (**b**).

**Figure 13 ijerph-19-07909-f013:**
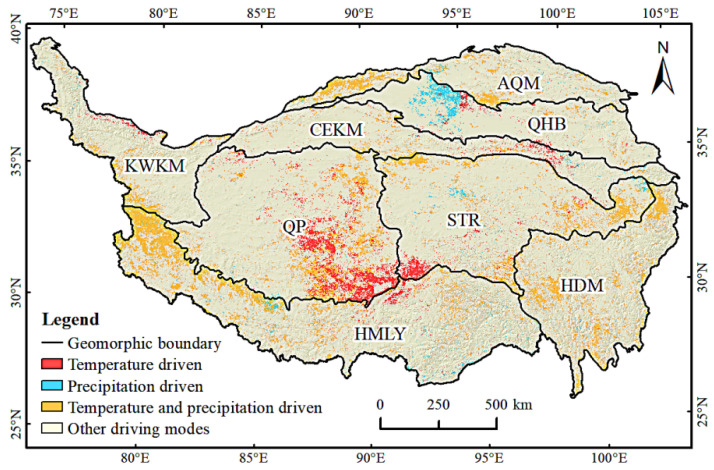
Climatic factor-driven zones.

**Figure 14 ijerph-19-07909-f014:**
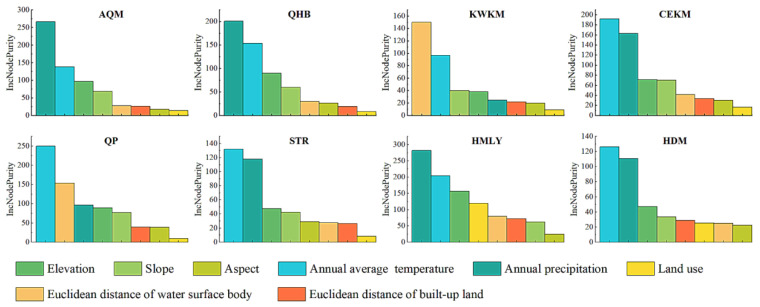
Ranking of the importance of factors in different geomorphological divisions of the QTP.

**Figure 15 ijerph-19-07909-f015:**
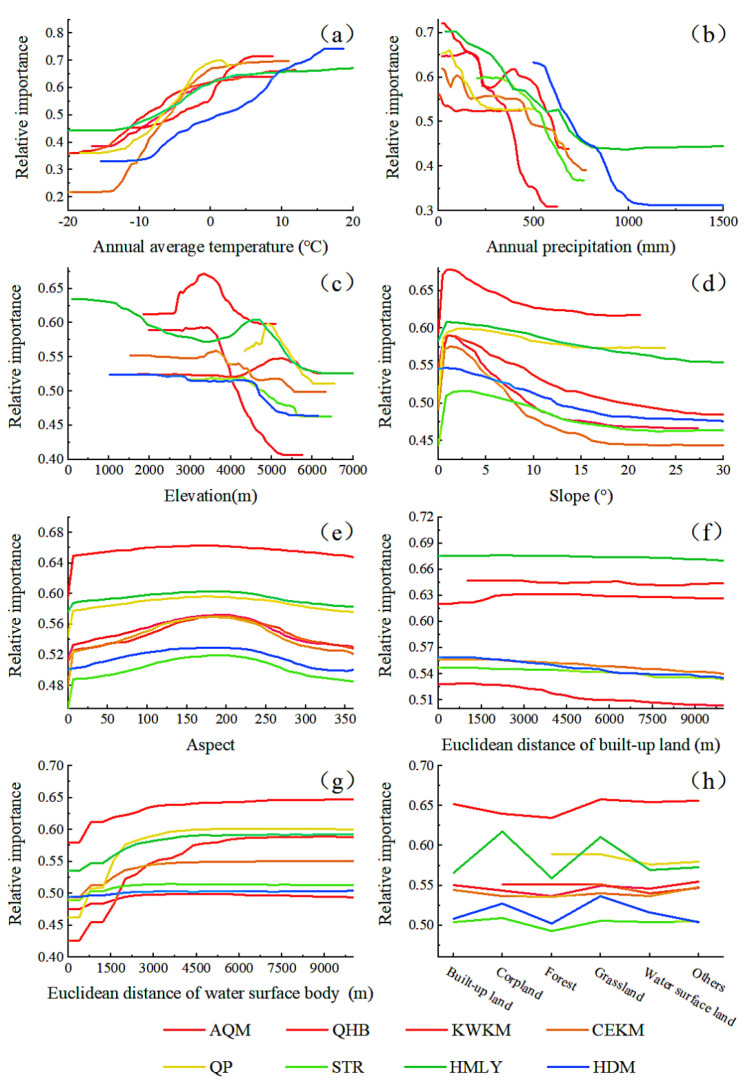
Relative importance of driving factors in different geomorphic regions. (**a**) Annual average temperature; (**b**) Annual precipitation; (**c**) Elevation; (**d**) Slope; (**e**) Aspect; (**f**) Euclidean distance of built-up land; (**g**) Euclidean distance of water surface body; (**h**) Land use and Land cover.

**Table 1 ijerph-19-07909-t001:** Data and data sources.

Data Class	Data	Data Sources	Spatial Resolution	Time of Data
Climatic factor	Precipitation	National Qinghai Tibet Plateau scientific data center (http://www.tpdc.ac.cn/zh-hans/, accessed on 7 April 2021)	1 km	2000–2017
	Temperature			2000–2017
Geomorphological factor	Elevation	China Geological Survey (https://www.cgs.gov.cn/, accessed on 21 April 2021)	1 km	2015
	Slope			
	Aspect			
Accessibility factor	Water surface body	National Qinghai Tibet Plateau scientific data center (http://www.tpdc.ac.cn/zh-hans/, accessed on 21 April 2021)	1 km	2015
	Built-up land			
Land use factor	LULC	National Qinghai Tibet Plateau scientific data center (http://www.tpdc.ac.cn/zh-hans/, accessed on 21 April 2021)	1 km	2015

**Table 2 ijerph-19-07909-t002:** Criteria for TVDI-driven zoning (It was reprinted with the permission from Ref. [56] Copyright 2022, Chen).

Types of TVDI Changes	Zoning Criteria		
	$r_{TVDI\;P,T}$	$r_{TVDI\;T,P}$	$R_{TVDI,\;TP}$
Precipitation driven	t ≥ t_0.05_		F ≥ F_0.05_
Temperature driven		t ≥ t_0.05_	F ≥ F_0.05_
Temperature and precipitation driven	t ≤ t_0.05_	t ≤ t_0.05_	F ≥ F_0.05_
Other drive types			F ≤ F_0.05_

**Table 3 ijerph-19-07909-t003:** Classification criteria of TVDI trend.

Grading Criteria		TVDI Trend
TVDI_Slope < 0	*p* > t_0.01_	Extremely significant decrease
	*p* ≥ t_0.05_ and *p* ≤ t_0.01_	Significant decrease
	*p* < t_0.05_	Non-significant decrease
TVDI_Slope > 0	*p* < t_0.05_	Non-significant increase
	*p* ≥ t _0.05_ and *p* ≤ t_0.01_	Significant increase
	*p* > t_0.01_	Extremely significant increase

**Table 4 ijerph-19-07909-t004:** TVDI classification criteria of drought in the QTP.

Classification of TVDI	[0, 0.2]	(0.2, 0.4]	(0.4, 0.6]	(0.6, 0.8]	(0.8, 1.0]
**Drought rank**	Extremely wet	Wet	Normal	Dry	Extremely dry
**Drought types**	No drought	No drought	No drought	Drought	Severe drought

**Table 5 ijerph-19-07909-t005:** Indicators of cluster analysis in each geomorphological division.

Geomorphological Division	TVDI-Max	TVDI-Min	TVDI-Mean	TVDI-SD	Slope-Mean	TVDI-t3	TVDI-t4	TVDI-t5	TVDI-t6
AQM	1.00	0.00	0.55	0.24	3.75	47.34%	44.11%	6.19%	2.37%
QHB	0.97	0.01	0.66	0.20	1.04	42.64%	51.48%	4.07%	1.82%
KWKM	1.00	0.00	0.55	0.18	1.09	20.64%	69.28%	8.21%	1.87%
CEKM	1.00	0.00	0.54	0.21	1.54	25.59%	62.20%	9.96%	2.25%
QP	0.99	0.00	0.59	0.15	5.94	46.76%	42.58%	6.96%	3.70%
STR	0.94	0.00	0.50	0.10	9.69	41.10%	46.45%	8.04%	4.41%
HMLY	1.00	0.03	0.59	0.18	−3.43	58.71%	37.09%	3.41%	0.79%
HDM	1.00	0.00	0.52	0.15	−1.83	73.79%	24.25%	1.57%	0.40%

## Data Availability

All data generated or analyzed during this study are included in this article.

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
