# Peer review of "Spatial-Temporal Evolution and Driving Forces of Drying Trends on the Qinghai-Tibet Plateau Based on Geomorphological Division"

_ijerph, 2022, doi:10.3390/ijerph19137909_

Round 1

Reviewer 1 Report

This study analyzed the spatial-temporal patterns of change by TVDI in the Qinghai-Tibet Plateau and the driving forces of drying trend including natural factors and human activities. Some minor revisions are required before it can be accepted, mainly on the improvement of its presentation. Its writing style needs to be further polished and largely improved.

Some detailed comments:

1.       Page 6, Figure 2: the figure is not very clear, please increase the resolution. In addition, why did you choose these 8 factors, and is there any scientific basis, such as what literature was referenced?

2.       Page 7, Table 1: The materials and data should be summarized and put into a table to introduce when the data was collected, the source, etc. to make it clearer. But in this table, the time of the data source is missing.

3.       Page 11: TVDI was divided into five grades. Please explain whether you are divided into 4 categories according to the data distribution histogram, or are there any classic literature classification. Please explain what size of the grid is being calculated in the process of TVDI and the driving forces.

Author Response

Point 1Page 6. Figure 2: the figure is not very clear, please increase the resolution. In addition, why did you choose these 8 factors, and is there any scientific basis, such as what literature was referenced?

Response 1: Thank you for this comment. Page 6. Figure 2: The picture has been replaced and the resolution of the picture has been improved.

Page 7, section 3.1.3: I have added detailed literature in the main text to further justify why these 8 factors were chosen, as follows:

Precipitation and temperature are important factors that affect drought variation, these two factors determine the spatial distribution of drought grade and drought frequency in the region [52]. Geomorphic factors lead to regional differentiation of drought, among which elevation plays a critical role in the spatial-temporal differentiation of drought spread in plateau mountains [53]. Human activities such as the increase in the domestic water supplies and the expansion of the urban areas can speed up the spread of drought [54]. Changes in land cover will affect the supply of atmospheric water, thus affecting the severity of regional drought [55]. Lakes, glaciers and other water areas will also affect the seasonal variation of regional drought [41]. So, we ……

The additions I have highlighted in red font in the paper.

Point 2Page 7, Table 1: The materials and data should be summarized and put into a table to introduce when the data was collected, the source, etc. to make it clearer. But in this table, the time of the data source is missing.

Response 2: Thank you for this comment. Page 7, table 1: The time represented by the data source has been added to the paper.

Point 3Page 11: TVDI was divided into five grades. Please explain whether you are divided into 4 categories accorcding to the data distribution histogram, or are there any classic literature classification. Please explain what size of the grid is being calculated in the process of TVDI and the driving forces.

Response 3: Thank you for this comment. Page 11, section 4.2: I have added literatures on TVDI classification. Such as the literature of [60-62]. TVDI was divided into five grades (Table 4): extremely wet (0≤TVDI≤0.2), wet (0.2<TVDI≤0.4), normal (0.4<TVDI≤0.6), dry (0.6< TVDI≤0.8), and extremely dry (0.8<TVDI≤1.0). Figure 6 and Figure 7 was also divied into five grades. When calculating the drought trend of TVDI, this paper uses 4 categories in statistics: non-significant decrease,non-significant increase, significant increase, and significant decrease.

The references I have added are highlighted in red font in the paper.

In section 3.1.1 and 3.1.3, , the spatial resolution of all data is stated to be 1km, so the calculated result and the driving factors is also 1km.

Point 4: Some minor revisions are required before it can be accepted, mainly on the improvement of its presentation.

Response 4: Thank you for this comment. I do my best to check the full text. All changes or additions are shown in red font in the manuscript. I will continue to work hard to overcome the language barrier in the writing of academic papers.

Reviewer 2 Report

This is an interesting study. Nevertheless, it needs some further improvements. In general, there are still some occasional grammar errors throughout the manuscript, especially the article "the," "a," and "an" is missing in many places; please make a spellchecking in addition to these minor issues. The reviewer has listed some specific comments that might help the authors further enhance the manuscript's quality.

  1. Specific Comments

·                  A list of acronyms is needed

Introduction

  • The objectives should be more explicitly stated.
  • Please elaborate on the introduction section. In this regard, the following literature may be helpful: DEM resolution effects on machine learning performance for flood probability mapping>>, you may consider additional references as well.
  • What is the novelty of this work?

  • Methods
  • The methodology limitation should be mentioned.
  • All variables should be explained.

  • Results
  • This section is well written.
  • Please improve the resolution of all figures, especially the text size.
  • Discussion
  • Overall, the discussion part is week. The Discussion should summarize the manuscript's main finding(s) in the context of the broader scientific literature and address any limitations of the study or results that conflict with other published work. 

Author Response

Response to Reviewer 2 Comments

Point 1: A list of acronyms.

Response 1: Thank you for this comment. The acronyms that appear in the paper are listed in Table A1. See Appendix A (Table A1) for details.

Point 2: The objectives should be more explicitly stated. What is the novelty of this work?

Response 2: Thank you for this comment. In section 1, this paper first introduces a simple definition of drought and its research methods, then puts forward the TVDI method and expounds its advantages and disadvantages, then discusses the drought research in the Qinghai Tibet Plateau from the two aspects of spatial-temporal characteristics and drought driving factors. Finally we explains the research methods and research purposes of this paper. The specific goals and the novelty of this paper are add in red font on Page 4, the details are as follows:

Therefore, the spatial-temporal differences and driving forces in the arid state of the Qinghai-Tibet Plateau are still worthy of attention. This paper attempts to: (1) analyze the feasibility of using the TVDI method to study drought in the Qinghai-Tibet Plateau, (2) clarify the spatial-temporal variation of drought in the Qinghai-Tibet Plateau, (3) evaluate the main driving forces affecting the Qinghai-Tibet Plateau and know the specific range of various factors that affect drought…….

The results of this paper provide a theoretical basis for the division of drought risk in the Qinghai-Tibet Plateau, the prevention of drought hazards and the sustainable development of regional economy.

The innovation of this paper is from two aspects. Firstly, the geomorphological division method was chosen to reduce the uncertainty of TVDI when calculating the drought status in large-scale areas. Secondly, eight factors in four categories, including meteorology, landform, land use and land cover (LULC), and human activities were selected to construct a random forest model. The main factors driving the drying trends in each geomorphological sub-division were explored and the contribution of each driving factor were ranked.

I have made further revisions in Section 1.

Point 3: The methodology limitation should be mentioned. All variables should be explained.

Response 3: Thank you for your comment. All formulas are checked again to ensure that all variables are explained in detail. Additionally, limitations of the TVDI method were added in Section 3.2.1. The text is as follows:

In areas with obvious geomorphological differences and large scale areas, there may be errors in using TVDI method to study drought. Therefore, this paper adopts the method of geomorphological divisions to reduce the impact of geomorphological factors on the results.

The additions I have highlighted in red font in the paper.

Point 4: Please improve the resolution of all figures, especially the text size.

Response 4: Thank you for this comment. I improved the resolution of all figures, and increased the size of the text in the figures. For example, Figure 5-Figure 9, Figure 11, Figure 13 and Figure 14.

All pictures can be arbitrarily enlarged by the editor, if required for publication.

Point 5: The discussion part is weak.

Response 5: Thank you for this comment.

The discussion part of this paper compares the previous research from three aspects, such as NDVI-LST feature space, the spatial-temporal and trend of TVDI, and the driving force of TVDI change, and emphasizes the inconsistencies between this paper and previous research. Although the discussion is based on a large number of literatures, there are still some drawbacks in the discussion part of this paper, so the content of this paper has been modified. The specific additions are as follows:

In Section 5.2, “In winter, under the combined influence of land-source water vapor and ocean-source water vapor, the central region of the QTP is obviously humid[67]. In addition, it is also affected by factors such as the decrease of temperature, vegetation dormancy, the expansion of snow cover area and the increase of snow depth [68].” is added.

In Section 5.3.3, ”Under the background of global warming and the imbalance of water towers in Asia, the glaciers on the QTP are in a state of continuous melting, their solid water is melting rapidly, and their liquid water is increasing[94]. This affects the hydrological process in the study area and relieves the drought pressure in some areas [95].” is added.

Point 6: In general, there are still some occasional grammar errors throughout the manuscript, especially the article "the," "a," and "an" is missing in many places

Response 6: Thank you for this comment. I will double check the full text. In the future I will continue to work hard to overcome language barriers in writing academic papers.

Notely, the modified text in this paper is shown in red font including the added references.
